# Exploring the Relationship between Urbanization and Ikization

Lü Ye [1], Yanguang Chen [2,*] and Yuqing Long [3]

1   College of Physical Education, Yangzhou University, Yangzhou 225002, China; 006738@yzu.edu.cn
2   Department of Geography, College of Urban and Environmental Sciences, Peking University,
    Beijing 100871, China
3   China Center for Urban Development, Beijing 100038, China; longyq@pku.edu.cn
*   Correspondence: chenyg@pku.edu.cn

**Abstract:** The phenomenon of Iks was first found by anthropologists and biologists, but it is actually a problem of human geography. However, it has not yet drawn extensive attention of geographers. Based on the relationship between urbanization and ikization, this paper is devoted to constructing a model to explain ikization. The research methods include literature-based analogy, mathematical modeling, empirical analysis, and numerical experiments. The main findings are as follows. First, the generalized production function can be used to model the behavior of ikization resulting from dramatic changes in the geographical environment and sudden cultural rupture. Second, nonlinear replacement dynamic models can be used to explain the possibility of rapid urbanization leading to ikization. Observational data is utilized to verify the fast urbanization mode, and numerical experimentation is employed to reveal the possible key factor causing ikization. The principal conclusions can be reached that social transition should adopt a relatively mild approach to changing, and protecting the geographical environment and reviving traditional culture contribute to national sustainable development.

**Keywords:** geographical environment; traditional culture; replacement dynamics; urbanization/urbanism; Ik effect; fractal dimension; Beijing

## 1. Introduction

Urbanization is a complex dynamic process based on growth of urban populations and migration of rural populations. A process of urbanization includes two nonlinear geographical processes: one is a gradual increase in the proportion between urban populations and total populations, and the other is societal changes to the people living in urbanized areas from a rural to an urban way of life [1,2]. The former is a quantitative change, and the latter involves qualitative change. The former indicates movement of rural populations to the urban regions and the concentration of urban people, while the latter is also known as urbanism, which is actually the characteristic way of interaction between urban inhabitants and the geographical environment [3]. For the urbanized people, there are two significant changes. One is the environmental change; that is, the rural environment is replaced by an urban environment. The other is cultural change; that is, the rural culture is substituted with urban culture.

As a whole, urbanization originates from industrialization. Without industrialization, there would be no urbanization in the modern sense [3]. For a long time, the character of urbanization is regarded as positive. It is seldom that people think of it as a double-edged sword under certain conditions because of nonlinear dynamics. In fact, a theory study has shown that fast urbanization can cause periodic oscillation or even chaos of the proportion of people living in urban areas [4]. The most terrible is the negative way in which urbanization can possibly give rise to ikization. The phenomenon of Iks was first found by anthropologists and biologists [5–8], but it is actually a problem of human geography [9]. However, it has not yet drawn extensive attention from geographers.

The replacement dynamics, a new theory of nonlinear systems, can be employed to explore the causality between ikization and fast urbanization. After all, urbanization is in essence a process of population replacement [10]. In this paper, a hypothesis of ikization is presented whereby a sudden and violent change in geographical environments results in the dismantling of traditional culture, which then results in collective depravity of a nationality. Based on this hypothesis, related mathematical models about the relationships between urbanization and ikization are constructed. The rest of the article is arranged as follows. In Section 2, a hypothesis of ikization will be put forward, and new models of nonlinear replacement dynamics are presented. In Section 3, the observational data of Beijing city will be utilized to testify the fast replacement models of urbanization, and numerical computation and simulation will be employed to reveal the key influence factor of the replacement process. In Section 4, several related questions will be discussed, and finally, in Section 5, the discussion will be concluded by outlining the main points of this study.

## 2. Theoretical Framework

### 2.1. A Hypothesis of Ikization

First of all, the concept of ikization should be illuminated. Ikization is a phenomenon of human geography, but it was first discussed in anthropology and biology instead of geography. Chen [9] coined the word ikization in terms of the small tribe of Iks in Uganda. The Ik people were formerly nomadic hunters and gatherers in the mountain valleys of northern Uganda, a country of east-central Africa. However, because the government decided to construct a national park, that is, Kidepo Valley National Park, they were compelled by law to give up hunting in the valleys and become farmers on poor hillside soil [7]. Since then, the tribe of Iks became a mean society [6]: "These people seem to be living together, clustered in small, dense villages, but they are really solitary, unrelated individuals with no evident use for each other. They talk, but only to make illtempered demands and cold refusals. They share nothing. They never sing. They turn the children out to forage as soon as they can walk, and desert the elders to starve whenever they can, and the foraging children snatch food from the mouths of the helpless elders . . . They breed without love or even casual regard. They defecate on each other's doorsteps. They watch their neighbors for signs of misfortune, and only then do they laugh." A British–American anthropologist, Colin Turnbull [7], found this tribe, and wrote a controversial book entitled "The Mountain People". This book interested scientists [6], but also attracted criticism from the author's peers [11]. Some scholars regarded that "this book cannot be discussed in any proper sociological terms, for we are provided with only snatches of data" [11,12]. Despite this objection, the book made the Ik people famous and become "*literary symbols for the ultimate fate of disheartened, heartless mankind at large*" [6].

The misfortune of the Ik people suggests that a tribe's ikization arises from a radical change in geographical environments and a break of traditional culture. Sudden and violent changes in geographical environments may induce discontinuity of the inherent culture. Cultural rupture will bring about a loss of people's sense of place and result in the collective depravity of a community or even a nationality. Actually, as indicated by the biologist, Thomas [6], "the Iks have transformed themselves into an irreversibly disagreeable collection of unattached, brutish creatures, totally selfish and loveless, in response to the dismantling of their traditional culture." There are two concepts that can be employed to describe the process of ikization: changes in the geographical environment and cultural breaking [5,8,9]. Both the environmental variation and cultural variation are involved with two aspects: one is absolute change, namely devastation, and the other is relative change, which is migration.

Now, a conceptual model can be built for our understanding the causality of ikization. If the degree of ikization is treated as a function (output variable, dependent variable, response variable), environmental variance and cultural break can be regarded as two argu-

ments (input variables, independent variables, explanatory variables). Thus the causality can be expressed as a linear regression equation as below:

$$Ikization = a + b_1 \times (environmental\ change) + b_2 \times (cultural\ break) \tag{1}$$

where $a$ refers to an intercept (constant), and $b_1$ and $b_2$ regression coefficients reflect the impact strength. Alternatively, the relationship between the cause and effect of ikization can also be described by a production function, which is a simple nonlinear regression equation, as follows:

$$Ikization = a \times (environmental\ change)^{b_1} \times (cultural\ break)^{b_1} \tag{2}$$

where $a$ denotes a constant, and $b_1$ and $b_2$ are two elasticity coefficients [13].

No human culture is independent of its geographical environment. Human beings depend on the earth surface, and culture results from the interaction and ecological relationships between human beings and the physical environment. In this sense, Equation (2) is more preferable than Equation (1), but Equation (1) is simpler and thus easier to understand than Equation (2). In fact, we can find a third function to model the causality of geographical processes and ikization, which will be displayed in Section 2.5.

### 2.2. Political and Economic Movements Resulting in Ikization

If we read the modern history of China, we can find that many events have brought about cultural catastrophes and environmental and ecological disasters. In some cases, a good motivation may lead to bad results. This implies the nonlinearity and thus complexity of social and economic systems. Today, the symptom of ikization has appeared among parts of the Chinese people after a long latent period, which suggests the effect of time lag between the causes and effects of a geographical process. Since the May Fourth Movement (1919), which was correlated with the New Culture Movement (1915–1923), Chinese traditional culture has been questioned, reconsidered, or even denied. This movement was regarded as an anti-imperialist and anti-feudal political and cultural movement and thought to be a historical progress of China, but its aftereffects are very complicated. During the Great Leap Forward and the People's Commune Movement (1958–1960), which represents the high point of ignorant mass folly, Chinese geographical circumstances and conditions as well as ecological systems suffered serious destruction [14]. Especially, the Great Proletarian Cultural Revolution (1966–1976) gave rise to both environmental and cultural devastation. Since the introduction of the policies of reform and opening-up at the end of 1978, and with the gradual establishment of a socialist market economic system from 1992, namely, after Deng's South Tour Speeches, China's national economy and cities developed rapidly [15]. However, this development once inflicted heavy losses of geographical environments due to the absence of sound rule of law (Table 1). In recent years, the Chinese government has finally realized the importance of protecting the geographical environment and revitalizing traditional culture. A series of new policies and measures have been introduced, which have significantly improved China's natural environment and ecological conditions. In particular, the value of traditional Chinese culture is also in the process of comprehensive reconsideration.

It is worth mentioning that political and economic corruption represents a significant factor contributing to collective ikization. In ancient China, corruption is an important ruling means used by almost all monarchs. If an emperor tried to plunder the national wealth through a state policy, he would be regarded as a bandit and would result in loss of the morale of nations. A foxy emperor never easily robbed the people of their wealth. Instead, he would rather utilize the corrupt officials to plunder the wealth of the people. If a corrupt official knew when to stop and just leave, he would not be punished in accordance with the law. However, if he was excessively greedy of gain and had suffered the bitter hatred of the people, he would be chastised in the name of the law. His house would be searched and his property, of course, the mammon of unrighteousness, would

be confiscated and finally obtained by the emperor. The emperor would kill two birds with one stone by punishing the corrupt officials and family members seriously. First, he acquired the wealth from his subjects by an indirect means. Second, he was regarded as a wise monarch and won the support of the people. For the emperor, the corrupt officials had two benefits. First, they were easily controlled because their handles were caught by the emperor. Second, they amassed huge wealth by plundering the national people for his monarch. What with the drawbacks of political institutions and what with the weakness of human nature, the corrupt officials emerged in an endless stream during the long process of Chinese history.

**Table 1.** Important historical events associated with great political and economic movements resulting in possible ikization of Chinese.

| Event | Time | Related Concept | Consequence |
|---|---|---|---|
| The May 4th Movement | 1919 | New Culture Movement | Reconsider Chinese traditional culture |
| Great Leap Forward | 1958–1960 | The People's Commune Movement | Environmental and ecological destruction |
| Great Cultural Revolution | 1966–1976 | Ten Chaotic Years | Environmental and cultural destruction |
| South Tour Speeches | 1992 | Reform and Opening-up | Geographical environmental destruction |
| Fast Urbanization | Recent years | House Demolition and City-Making Movement | Geographical environmental destruction |

In modern society, the source of political corruption is the very source of unsustainable development. The ancient emperors did not care about official corruption, which, as indicated above, was an approach for the emperors to make a fortune. After all, the officials could not emigrate abroad, so they would not transfer the ill-gotten gains overseas. What is more, due to the technical level, the extent of damage to the geographical environments caused by the official corruption was small. However, today, the situation is different from that in ancient times. In order to acquire an immense amount of treasure through exploitation of natural resources, the corrupt officials always undermine environments crazily by colluding with merchants. Malfeasant official behaviors rely heavily on gaudy approaches such as reservoir resettlement and fast urbanization. Because of the advanced technology, it is easy for officers and businessmen to destroy the natural environments. In particular, they will not be responsible for the land as both the corrupt officials and illegal dealers can immigrate overseas. In a sense, modern corruption can cause great harm to the geographical environments and the relations between man and land, which lead to ikization of people, which in turn aggravate official corruption. Therewith a vicious circle comes into being, harming the country and bringing disaster to the people (Figure 1).

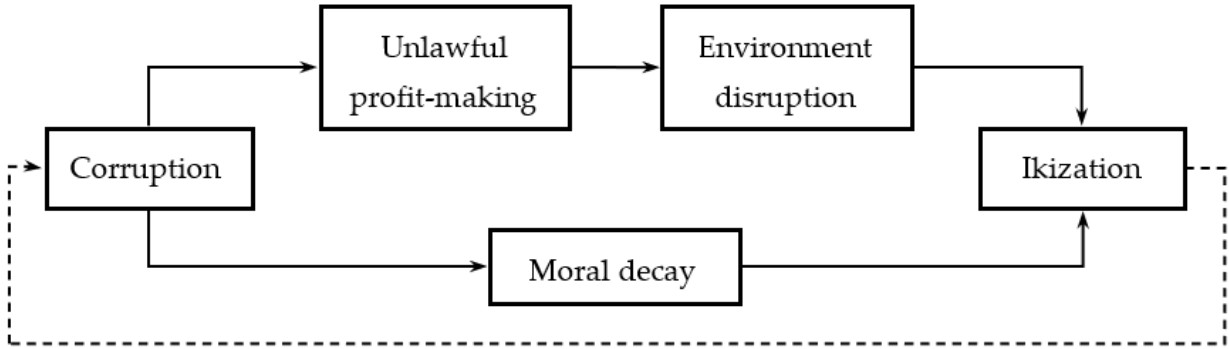

**Figure 1.** The causal sequence from official corruption to ikization and in turn to corruption.

### 2.3. Urbanization and Ikization

Urbanization may be one of the significant factors of ikization in the history of mankind. Urbanization implies the process of migration of rural populations into cities

and towns, which results in an increasing proportion of the regional population residing in urban settlements. A city and a network of cities are complex spatial systems [16–20], and urbanization is a self-organizing process of urban evolution [21–24]. In a sense, complex spatial patterns and self-organized processes represent two different sides of the same coin of city development. The dynamics of urbanization can be understood by two processes. First, urbanization indicates a kind of phase transition from a rural to an urban settlement system [25,26]. Phase transition is originally a physical term, which is often associated with the concept of self-organized criticality (SOC) [22,27]. In physics, a phase transition denotes the transformation of a thermodynamic system from one state (e.g., liquid) of matter to another one (e.g., solid or gaseous) by heat transfer [28,29]. Today, the term can be employed to describe an evolution of a geographical system from a rural state into an urban state by urbanization. Second, urbanization suggests complex replacement dynamics: rural settlements are replaced by urban settlements, and the rural population is replaced by the urban population [10,30,31]. In essence, phase transition and replacement dynamics also represent two different sides of the same coin of urbanization.

In geography, urbanization involves both quality and quantity. A common knowledge is that the process of urbanization involves four aspects: urban system, urban form, urban ecology, and urbanism [1,3]. Both urban system and urban form can be well described from the prospective of natural cities [32,33]. However, it is difficult to study urban ecology and urbanism through modern technology. Among the four aspects, urbanism reflects the quality of urbanization and is more correlated with ikization. The term "urbanism" suggests that the culture or way of rural life of the urbanized people is substituted with the urban culture or way of urban life. The ikization can be associated with urbanization because the transfer of populations from rural regions to urban regions may give rise to abrupt changes in geographical environments and traditional culture.

For the city dwellers, the phase transition of urbanization includes two replacement processes during rural–urban population migration. First, the rural geographical environment is replaced by an urban geographical environment. Second, the rural cultural way of life is replaced by an urban cultural way of life. Thus the original sense of place will be lost. It will take a long time for the migratory dwellers to form a new sense of place. This may result in ikization (Figure 2). In fact, Thomas [6] once pointed out: "Cities have all the Ik characteristics. They defecate on doorsteps, in rivers and lakes, their own or anyone else's. They leave rubbish. They detest all neighboring cities, give nothing away. They even build institutions for deserting elders out of sight." This suggests that urban inhabitants look similar to Iks because of environmental and cultural changes.

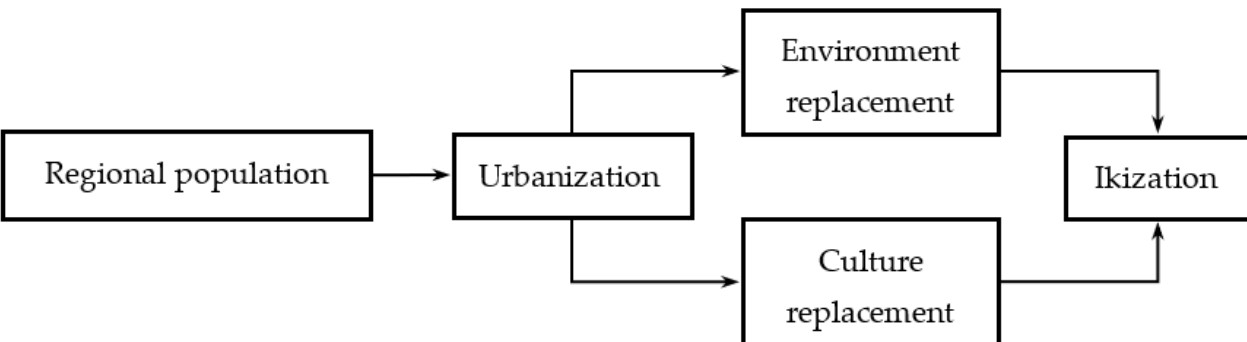

**Figure 2.** The replacement dynamics of urbanization which may result in ikization.

It is rapid urbanization instead of natural urbanization that results in ikization of a community or even a nationality. In China, ikization may proceed from fast urbanization because of large-scale demolition of traditional urban communities and rural settlements. A new word, chaiqian, has emerged as the times require. Chaiqian means removal and relocation, that is, dismantlement of houses and movement of local residents. In many regions, government officials and business owners acted in collusion with each other to

engage in rent-seeking of land. Many traditional settlements were pulled down, and local residents were compelled to move to the appointed places and were relocated. Many people became landless peasants coming between the urban state and rural state, forming a kind of marginal people living at the edge of modern society. Their situations were similar to Iks. Overnight, their way of life was thoroughly remodeled. In short, their living environments were suddenly changed, and their inherent culture was broken down. They felt that they were rapidly transferred from one time and space to another time and space.

### *2.4. Logistic Replacement*

The process of urbanization is associated with spatial replacement dynamics. Replacements are ubiquitous phenomena in both nature and society, which always take on a sigmoid curve [30,34–38]. In the simplest case, the replacement dynamics can be modeled with a logistic function. Among various replacement processes, urbanization and urban growth are two kinds of typical geographical substitution. Urbanization is a process of urban–rural population replacement, which has been studied for many years [10,31]. A new finding is that urban growth represents the complex dynamics of spatial replacement, which can be modeled with Boltzmann's equation and logistic function [30]. Urban replacement is one of the nonlinear processes of geographical replacement. More importantly, geo-replacement is where the natural environment is replaced by human systems by degrees. In geography, the very basic and significant topic is the nonlinear relation between man (human societies) and land (natural environment) [39]. The man–land relation can be regarded as the most important ecological relation in this world. Because of the interaction between human being and the surface of Earth, the primary productivity or even the secondary productivity are gradually consumed by humankind. So far, man has used up more than 40% of the primary productivity. In other words, human beings have transformed the first nature into the second nature and the third nature [10,40]. To characterize this replacement process, a new measurement, the ratio of the primary productivity to the total productivity in a region, $p(t)$, can be defined by:

$$p(t) = \frac{y(t)}{x(t) + y(t)} \times 100\% \tag{3}$$

where $x$ represents the primary productivity, and $y$ is the other productivity. Accordingly, the ratio of the other productivity to the primary productivity (POR) can be defined as $o(t) = y(t)/x(t)$ [10]. Based on Equation (3), a logistic model of man–land replacement can be built, and its mathematical expression is as below:

$$p(t) = \frac{1}{1 + (1/p_0 - 1)e^{-kt}} \tag{4}$$

in which $p_0$ is the initial value of the productivity ratio $p(t)$, i.e., the ratio of productivity of time $t = 0$, and $k$ is the original rate of growth.

Because of population explosion, natural space has been rapidly replaced by human space all over the world. In fact, the human race depletes geographical environment so fast by predatory exploitation of natural resources that the man–land relation becomes unstable. If we look at the earth from space, we will be surprised at its change. In particular, China as seen from the satellite is similar to a piece of withered and yellow leaf rather than a dark green leaf [41]. We cannot find a place in our country as a Land of Peach Blossoms, i.e., a land of idyllic beauty. If things continue this way, periodic oscillations or even chaos may arise some day in the future [10]. If the ratio of the other productivity to the total productivity is substituted by the level of urbanization, $L(t)$, we will have a logistic model of urban–rural replacement, that is:

$$L(t) = \frac{1}{1 + (1/L_0 - 1)e^{-kt}} \tag{5}$$

where $L_0$ is the initial value of the urbanization level $L(t)$ at the time $t = 0$, and $k$ is the inherent rate of growth of the urban population proportion.

### 2.5. Step Replacement and Ikization

Logistic replacement is a natural replacement, which cannot result in ikization. A fast urbanization curve should be described with a quadratic logistic function, and can be termed quadratic logistic replacement. Both logistic replacement and the quadratic logistic function belong to sigmoid replacement. However, another type of geographical replacement, step replacement, will give rise to ikization. The so-called ikization is in fact a process of human replacement: noble people will be substituted by mean people (Figure 2). The step replacement is a dynamic process that can be characterized by the unit-step function

$$p(t) = \begin{cases} 0, & t < t_0 \\ 1, & t \geq t_0 \end{cases} \tag{6}$$

where $t_0$ denotes a critical time. The unit-step function can be treated as the extreme special case of the logistic function, or the logistic function can be thought of as a smooth approximation of the unit-step function. Actually, in Equation (4), if $t = 0$, we will have $p(t) = p_0 = 0$, while if $t \rightarrow \infty$, then it follows $p(t) = 1$. Therefore, this replacement can also be termed 0–1 replacement. The unit-step function can be employed to model the processes of substitution such as reservoir resettlement and large-scale dismantlement of houses and movement of local residents. For urbanization, three types of replacement dynamics can be tabulated and illustrated as below (Table 2, Figure 3). During urbanization, the original sense of place of the urbanized people gets lost, and it will take a long time for the migratory dwellers to form a new sense of place. The loss of place sense may be one of influencing factors of ikization.

**Table 2.** Three types of urbanization replacement and the corresponding mathematical models.

| Urbanization | Replacement | Model | Feature |
|---|---|---|---|
| Natural urbanization | Natural replacement | Logistic function | Gradual change |
| Fast urbanization | Fast replacement | Quadratic logistic function | Rapid change |
| Dismantlement/movement | Step replacement | Unit-step function | Sudden change |

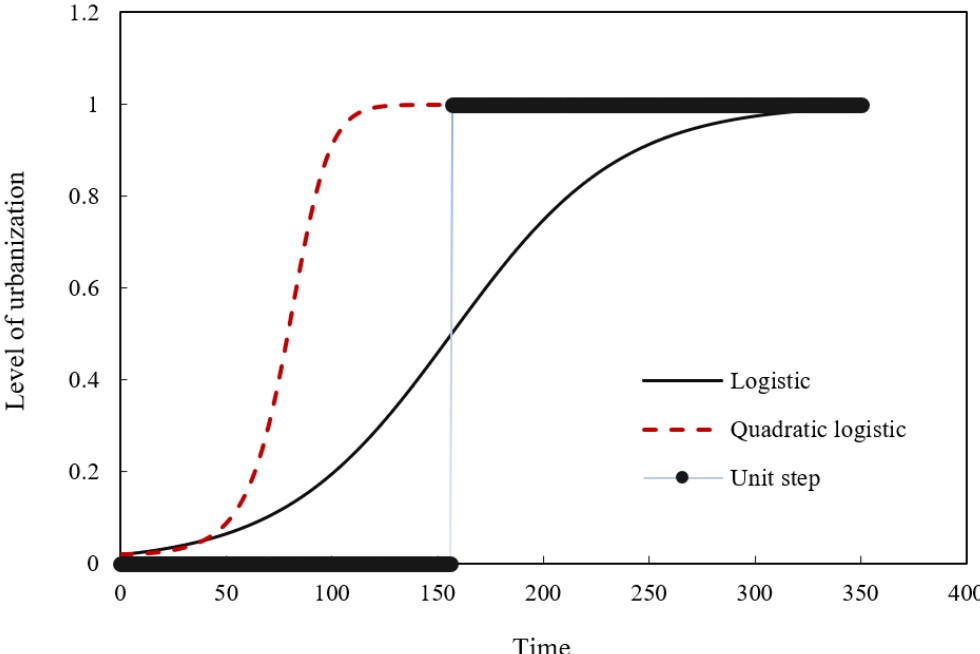

**Figure 3.** Three types of urbanization curves indicating three types of replacement dynamics.

In theory, ikization can be measured with 0 and 1. If ikization appears, it is 1, or else it is 0. Thus the ikization process can be depicted with a unit-step function, Equation (6). Based on the step replacement, the conceptual model of ikization can be substituted with a logistic regression model in the following form:

$$Ikization = \frac{1}{1 + ae^{-[b_1 \times (environmental\ change) + b_2 \times (cultural\ break)]}} \tag{7}$$

where $a$, $b_1$, and $b_2$ are parameters. In this instance, the degree of ikization will be measured with a dummy variable rather than a metric variable. A dummy variable is also known as a binary variable, Boolean indicator, categorical variable, design variable, indicator variable, or qualitative variable, which is always represented by 0 and 1 [42,43]. Now, we have three possible mathematical models to describe the causality of ikization (Table 3).

**Table 3.** Three possible conceptual models of ikization indicative of causes and effect.

| Relation | Ikization Measure | Function | Model |
|---|---|---|---|
| Linear relation | Metric variable | Linear function | $y = a + b_1 x_1 + b_2 x_2$ |
| Nonlinear relation | Metric variable | Production function | $y = a x_1^{b_1} x_2^{b_2}$ |
| | Dummy variable | Logistic function | $y = \frac{1}{1 + ae^{-(b_1 x_1 + b_2 x_2)}}$ |

Note: In the table, $y$ denotes the degree of ikization, $x_1$ refers to environmental change, and $x_2$ to cultural break. As indicated above, $a$, $b_1$, and $b_2$ are three parameters. Linear process cannot lead to complexity; the possible models for describing and explaining ikization are nonlinear functions.

## 3. Methods and Results

### 3.1. Materials and Methodology

The theoretical framework of the relationship between urbanization and ikization was outlined above. Next, two aspects of investigation need to be carried out. One is to use data to demonstrate that the replacement process satisfies a logistic process or a generalized logistic process, and the other is to prove that excessive replacement can lead to abnormal results. To achieve this, it is necessary to use certain scientific methods. In fact, there were three ways now to proceed in science: mathematical theory, laboratory experiment, and computer modeling [44]. This study cannot rely heavily on laboratory experiments, but mathematical models and computer simulation can be used to illustrate the problem. The city of Beijing can be taken to make a positive analysis, and a model of nonlinear dynamics based on Equations (4) and (6) can be employed to make numerical calculations and simulation analysis.

The basic tools of replacement dynamics including squashing functions, sigmoid curves, and allometric equations, can be utilized to analyze the rapid growth of Beijing city. The variables comprise the traditional measures, urban area and population size, and a new measurement, the scaling exponent. However, a city has no characteristic scale and urban form cannot be effectively described with common measures such as area and size [17]. Urban area can be subjectively defined rather than objectively measured. As an alternative, urban form and growth should be described with the fractal dimension. The fractal dimension is a kind of scaling exponent indicating the extent of space filling and the degree of spatial complexity. The fractal dimension can be evaluated with the box-counting method, and the fractal dimension growth can be modeled with sigmoid functions [30]. Although the urban area depends on definition rather than objective measurement, it is necessary to define a comparable urban boundary before calculating the fractal dimension in order to ensure the comparability of calculation results. At present, three approaches can be employed to designate urbanized areas or demarcate urban agglomerations. The first is the city-clustering algorithm (CCA) presented by Rozenfeld et al. [45,46], the second is the fractal-based method presented by Tannier et al. [47], and the third is a variant of the CCA based on street nodes/blocks developed by Jiang and Jia [48]. In this work, the urbanized area of Beijing is determined by using the CCA. By means of 15 years of remote sensing



data of the urbanized area from 1984 to 2018, we can examine the urban sprawl of Beijing in the past 30 years.

### 3.2. Empirical Results of Beijing

Because of the destruction of the geographical environment and the rupture of traditional culture, the symptom of ikization seems to have appeared partially in Chinese people. Various phenomena of Iks such as corruption, indifference, schadenfreude, no sense of responsibility, and making counterfeit goods have been discussed in detail [9]. On the other hand, the problems of fast urbanization of China have been studied by many scholars from varied perspectives. However, the mathematical modeling and quantitative analyses are seldom reported on rapid urbanization in the literature. In China, the typical signs of fast urbanization are urban sprawl, bubble economy of real estate, and large-scale demolition of traditional settlements. These signs are correlated with one another. As indicated above, one of the aspects of urbanization is urban form [1,3], which takes on sprawl during the stage of rapid urbanization. Due to scale dependence, the fractal dimension becomes an effective parameter for characterizing urban morphology [17,30]. Beijing, the capital of China, can be regarded as a microcosm of China, which in turn can be treated as a macrocosm of Beijing. The model of a city's growth is always consistent with the model of urbanization of its country. We might as well take Beijing as an example to illustrate the dynamic process of China's fast urbanization by means of fractal parameters. The calculation results of fractal parameters of Beijing's urban form for 15 years are as follows, and the time span of the data is 35 years (Table 4).

**Table 4.** The capacity dimension, information dimension, and correlation dimension of Beijing city (1984–2018).

| Year | Time | Capacity Dimension | Goodness-of-Fit | Information Dimension | Goodness-of-Fit | Correlation Dimension | Goodness-of-Fit |
|------|------|--------------------|-----------------|-----------------------|-----------------|-----------------------|-----------------|
| $n$ | $t$ | $D_0$ | $R^2$ | $D_1$ | $R^2$ | $D_2$ | $R^2$ |
| 1984 | 0 | 1.7412 | 0.9908 | 1.6189 | 0.9978 | 1.5701 | 0.9898 |
| 1988 | 4 | 1.7381 | 0.9937 | 1.6328 | 0.9958 | 1.5975 | 0.9884 |
| 1989 | 5 | 1.7921 | 0.9933 | 1.6889 | 0.9994 | 1.6453 | 0.9950 |
| 1991 | 7 | 1.7979 | 0.9941 | 1.6967 | 0.9986 | 1.6594 | 0.9939 |
| 1992 | 8 | 1.7853 | 0.9957 | 1.6925 | 0.9985 | 1.6605 | 0.9947 |
| 1994 | 10 | 1.8313 | 0.9958 | 1.7600 | 0.9998 | 1.7268 | 0.9977 |
| 1995 | 11 | 1.8527 | 0.9960 | 1.7664 | 0.9996 | 1.7312 | 0.9969 |
| 1996 | 12 | 1.8470 | 0.9967 | 1.7634 | 0.9994 | 1.7296 | 0.9965 |
| 1998 | 14 | 1.8501 | 0.9967 | 1.7686 | 0.9995 | 1.7361 | 0.9971 |
| 1999 | 15 | 1.8718 | 0.9973 | 1.7983 | 0.9996 | 1.7706 | 0.9978 |
| 2001 | 17 | 1.8880 | 0.9978 | 1.8195 | 0.9998 | 1.7926 | 0.9985 |
| 2006 | 22 | 1.9221 | 0.9987 | 1.8677 | 0.9999 | 1.8445 | 0.9997 |
| 2009 | 25 | 1.9367 | 0.9991 | 1.8905 | 0.9999 | 1.8711 | 0.9999 |
| 2014 | 30 | 1.9754 | 0.9997 | 1.9272 | 0.9999 | 1.9068 | 1.0000 |
| 2018 | 34 | 1.9823 | 0.9998 | 1.9386 | 0.9999 | 1.9194 | 1.0000 |

Note: The original data sources are satellite remote sensing images with a spatial resolution of 30 m, including TM data from the US Land Resources Satellite and CCD data from domestic environmental satellites. The interpretation method is supervised classification, with a classification accuracy of over 86%. See File S1 in Supplementary Materials.

Fractal dimension increase takes on a squashing effect and can be modeled by squashing functions. The squashing functions include the conventional logistic function, fractional logistic function, and quadratic logistic function [49]. The fractal parameters of Chinese urban growth can be described with a quadratic logistic function [49,50]. The curve-fitting method and least squares method can be utilized to estimate the parameters of models (Table 5). There are three basic and important parameters in a multifractal spectrum; that is, capacity dimension, information dimension, and correlation dimension. Based on the

dataset consisting of 15 data points from 1984 to 2018, a quadratic logistic model of Beijing's urban growth based on capacity dimension can be constructed as below:

$$\hat{D}_0(t) = \frac{1.9925}{1 + 0.1286e^{-(0.0529t)^2}} \tag{8}$$

where $t$ denotes time, and $D_0(t)$ refers to the capacity dimension of time $t$. The goodness-of-fit is about $R^2 = 0.9835$ (Figure 4a). In terms of this model, the inherent rate of growth of capacity dimension is approximately 0.0529.

**Table 5.** Quadratic logistic models parameters and statistics based on the capacity dimension, information dimension, and correlation dimension of Beijing city.

| Parameter | Capacity Dimension $D_0$ | Information Dimension $D_1$ | Correlation Dimension $D_2$ |
|---|---|---|---|
| Order of time | 2 | 2 | 2 |
| Fractal dimension capacity $D_{max}$ | 1.9925 | 1.9485 | 1.9294 |
| Inherent growth rate $k$ | 0.0529 | 0.0550 | 0.0556 |
| Coefficient $D_{max}/D_0$-1 | 0.1286 | 0.1743 | 0.1920 |
| Goodness-of-fit $R^2$ | 0.9835 | 0.9902 | 0.9918 |

Capacity dimension is based on the dummy variable and reflects space-filling extent. It cannot mirror detailed features of spatial distribution. The information dimension is based on metric variables and can be utilized to reveal spatial uniformity. Spatial uniformity and spatial heterogeneity are two different sides of the same coin. Where Beijing is concerned, based on the dataset comprising 15 data points of the information dimension, a quadratic logistic model can be made as follows:

$$\hat{D}_1(t) = \frac{1.9485}{1 + 0.1743e^{-(0.0550t)^2}} \tag{9}$$

in which $D_1(t)$ denotes the information dimension of time $t$. The goodness-of-fit is about $R^2 = 0.9902$ (Figure 4b). In light of this model, the inherent growth rate of the information dimension is around 0.0550. The relative growth rate of the capacity dimension (0.0529) is slightly less than that of the information dimension (0.0550). This suggests that the spatial homogenization of urban morphology in Beijing is faster than the spatial filling.

The correlation dimension is direct measure of spatial complexity in geographical systems. If we want to investigate the spatial dependence and complication of urban morphology, it is necessary to build a model with the help of the correlation dimension. As far as Beijing is concerned, based on the dataset comprising 15 data points of the correlation dimension, a quadratic logistic model can be given as below:

$$\hat{D}_2(t) = \frac{1.9294}{1 + 0.1920e^{-(0.0556t)^2}} \tag{10}$$

in which $D_2(t)$ denotes the correlation dimension of time $t$. The goodness-of-fit is about $R^2 = 0.9918$ (Figure 4c). According to this model, the inherent growth rate of the correlation dimension is around 0.0556. Obviously, the relative growth rate of the information dimension (0.0550) is slightly less than that of the correlation dimension (0.0556). This indicates that the spatial correlation speed of urban morphology in Beijing is higher than the speed of spatial homogenization, and even higher than the speed of spatial filling.

As indicated above, a city's growth is consistent with urbanization of a nation. The natural urbanization curve can be described with the conventional logistic function. If the urbanization curve exhibits a quadratic logistic function, the latent scaling exponent, i.e., the order of time, is 2, and thus it indicates fast urbanization. After all, the quadratic logistic growth comes between exponential growth and the conventional logistic growth.

The quadratic logistic model suggests that the relative growth rate of urban sprawl is much higher than the relative growth rate under normal circumstances. The model form of fractal dimension growth of Beijing's urban form lends further support to the suggestion of fast urbanization of China.

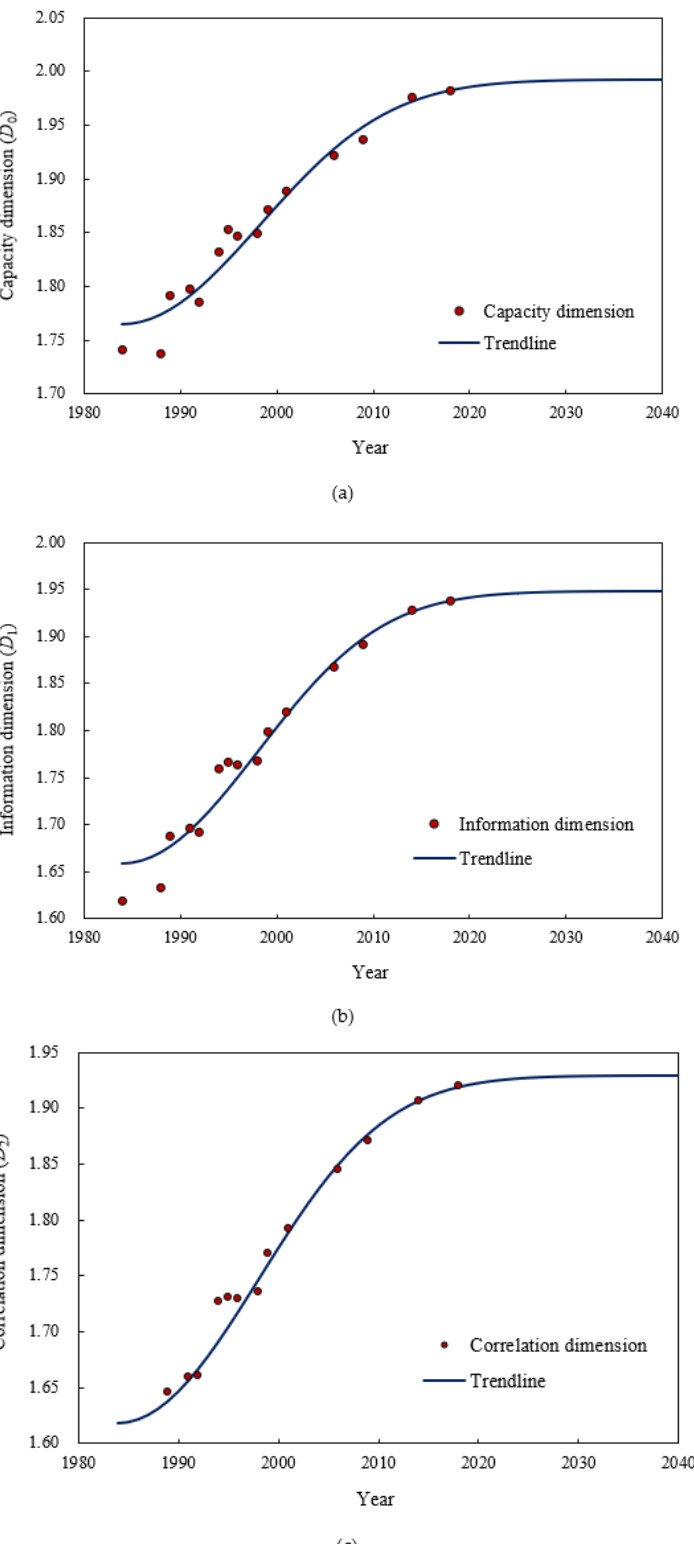

**Figure 4.** The quadratic logistic curves of urban growth of Beijing city based on fractal parameters (1984–2040). The model form and order of time suggests a rapid replacement and fast urbanization process. (**a**) Capacity dimension; (**b**) Information dimension; (**c**) Correlation dimension.

Quantitative and modeling analysis includes two main aspects: one is the mathematical structure of the model, and the other is the model parameters and their changes. Now, based on the mathematical model structure and parameters, we can analyze the basic properties of Beijing's urbanization. Where model structure is concerned, the curves of fractal dimension growth of Beijing's urban form can be modeled by the quadratic logistic function rather than the common logistic function. As indicated above, a city's growth is consistent with urbanization of a nation. The natural urbanization curve can be described with the conventional logistic function. If the urbanization curve exhibits a quadratic logistic function, the latent scaling exponent, i.e., the order of time, is 2, and thus it indicates a fast urbanization process. After all, the quadratic logistic growth comes between exponential growth and the conventional logistic growth. The quadratic logistic model suggests that the relative growth rate of urban sprawl is much higher than the relative growth rate under normal circumstances. The model structure of fractal dimension growth of Beijing's urban form lends further support to the suggestion of fast urbanization of China.

A model's form reflects the macroscopic structure of a system, while the model parameters reflect the element relationships at the microscopic level of the system. As far as the model's parameters are concerned, three values should be investigated. Firstly, the capacity parameters of the fractal dimension, $D_{max}$, are all greater than $D = 1.92$, approaching the limit of the Euclidean dimension of the embedding space, $d = 2$. This implies that, after the city of Beijing reaches its limit of growth, the buffer space of this city will be very small. Secondly, although the inherent growth rates of fractal parameters are relatively close to one another, there are subtle differences between them. As mentioned above, the inherent growth rate of the correlation dimension is higher than that of the information dimension, and the inherent growth rate of the information dimension is higher than that of the capacity dimension. This means that the multifractal structure in Beijing is trending towards a monofractal structure. Considering the very high capacity value of fractal parameters ($D_{max}$), we judge that the fractal structure of Beijing may even degenerate towards Euclidean geometric structure. Thirdly, the order of the time is 2, suggesting fast urban growth. The order of time is actually a latent scaling exponent, which comes between 1 and 2, in theory. The higher the value, the faster the urban growth. The theoretical upper limit is 2. The time order of Beijing's urban fractal dimension growth has reached its theoretical limit. As we know, Beijing is a megacity in the world. If the characters of fast urbanization appear in the middle-sized and small cities, it can be regarded as normal. However, a megacity such as Beijing shows varied signs of fast urbanization, and so it is abnormal. This suggests a sort of urbanization bubble or bubble economy behind urbanization, which in turn suggests environmental disruption and cultural rupture.

### 3.3. Numerical Calculation and Simulation

The above empirical analysis gives the observational evidence of the fast replacement process and one of its mathematical models. The key influencing factor of replacement dynamics can be revealed by numerical computation and simulation. Based on Equation (3), a simple model of the nonlinear replacement process can be expressed as a pair of differentiation equations as below [10]:

$$\begin{cases} \frac{dx(t)}{dt} = ax(t) - \frac{bx(t)y(t)}{x(t)+y(t)} \\ \frac{dy(t)}{dt} = cy(t) + \frac{dx(t)y(t)}{x(t)+y(t)} \end{cases} \quad (11)$$

where the meanings of symbols $x$ and $y$ are the same as those in Equation (3), and $a$, $b$, $c$, $d$ are four parameters (Table 6). From Equation (11), a logistic function indicating a replacement curve such as in Equation (4) can be derived. This suggests that Equations (4) and (5) are based on the nonlinear dynamics model, Equation (11). However, Equation (11) is a pair of differentiation equations, which can be applied to mathematical reasoning rather

than numerical experimentation. Numerical experiments can be conducted using difference equations. Discretizing Equation (11) yields a pair of difference equations as follows:

$$\begin{cases} x(t+1) = (1+a)x(t) - \frac{bx(t)y(t)}{x(t)+y(t)} \\ y(t+1) = (1+c)y(t) + \frac{dx(t)y(t)}{x(t)+y(t)} \end{cases} \tag{12}$$

which is a discrete replacement dynamical model and can be employed to make numerical computation and simulation.

**Table 6.** The meaning of model parameters of replacement dynamics based on urban–rural population interaction.

| Type | Parameter | Meaning | Value |
|---|---|---|---|
| Rural population | $a$ | Rural natural growth rate | 0.375 |
| | $b$ | Rural-urban coupling strength | 2, 2.65, 2.95, 3.25 |
| Urban population | $c$ | Urban natural growth rate | 2, 2.65, 2.95, 3.25 |
| | $d$ | Urban-rural coupling strength | 0.005 |

Note: This table takes the urban–rural population interaction model as an example to illustrate the meaning of the parameters in the replacement dynamics model. See File S2 in Supplementary Materials.

Adjusting model parameter values will output different numerical calculation results. Fix some parameters, change a certain parameter, and see if there are significant differences in the output results. Through such operations, it is possible to distinguish between primary and secondary influencing factors. This process is similar to laboratory experiments, so it can be considered as a numerical simulation or mathematical experiment [10]. The results show that the key influencing factors are the coupling parameters, $b$ and $d$, rather than the natural growth parameters, $a$ and $c$. In theory, $b$ equals $d$. The values of $b$ and $d$ reflects the strength of interaction and speed of replacement. When the values of $b$ and $d$ are small, the replacement process displays a logistic curve (Figure 5a). As the parameter values of $b$ and $d$ increase, the substitution process begins to fluctuate and periodic oscillation occurs (Figure 5b). The higher the parameter values of $b$ and $d$, the more complicated the periodic oscillation becomes (Figure 5c). When the parameter value exceeds a certain threshold, a complex oscillation without any period will emerge, and this phenomenon is called chaotic state (Figure 5d). This means that if the replacement process is too fast, the evolution process will become unstable, and ultimately lead to uncertain evolution results. Chaos indicates disorder. If social evolution becomes out of order, unpredictable things will happen and adverse phenomena will appear. It is not surprising that the replacement of human and environmental spaces or the rapid replacement of urban and rural populations and culture leads to ikization.

There is always a time-lag relation between causes and effects in the evolution of complex systems. The past causes lead to the present and future effects, and the present causes lead to the future effects. Where there is time lag, there is nonlinearity; and where there is nonlinearity, there is complexity. Replacement dynamics involve time lag. The madness of men not only brings about the revenge of nature on human beings but also results in the fall of men themselves. The large-scale demolition of traditional urban communities and rural settlements in the process of fast urbanization suggests more adverse outcomes related with ikization in the future. It is necessary to protect the geographical environment against disruption, and to inherit and develop traditional culture in order to avoid ikization of a nationality. One of the important approaches to solving the problems caused by fast urbanization is to safeguard and reconstruct the geographical environment so that rural situations and culture will be naturally replaced by urban situations and culture. Fortunately, the Chinese government has realized the problems and taken positive measures to solve them.

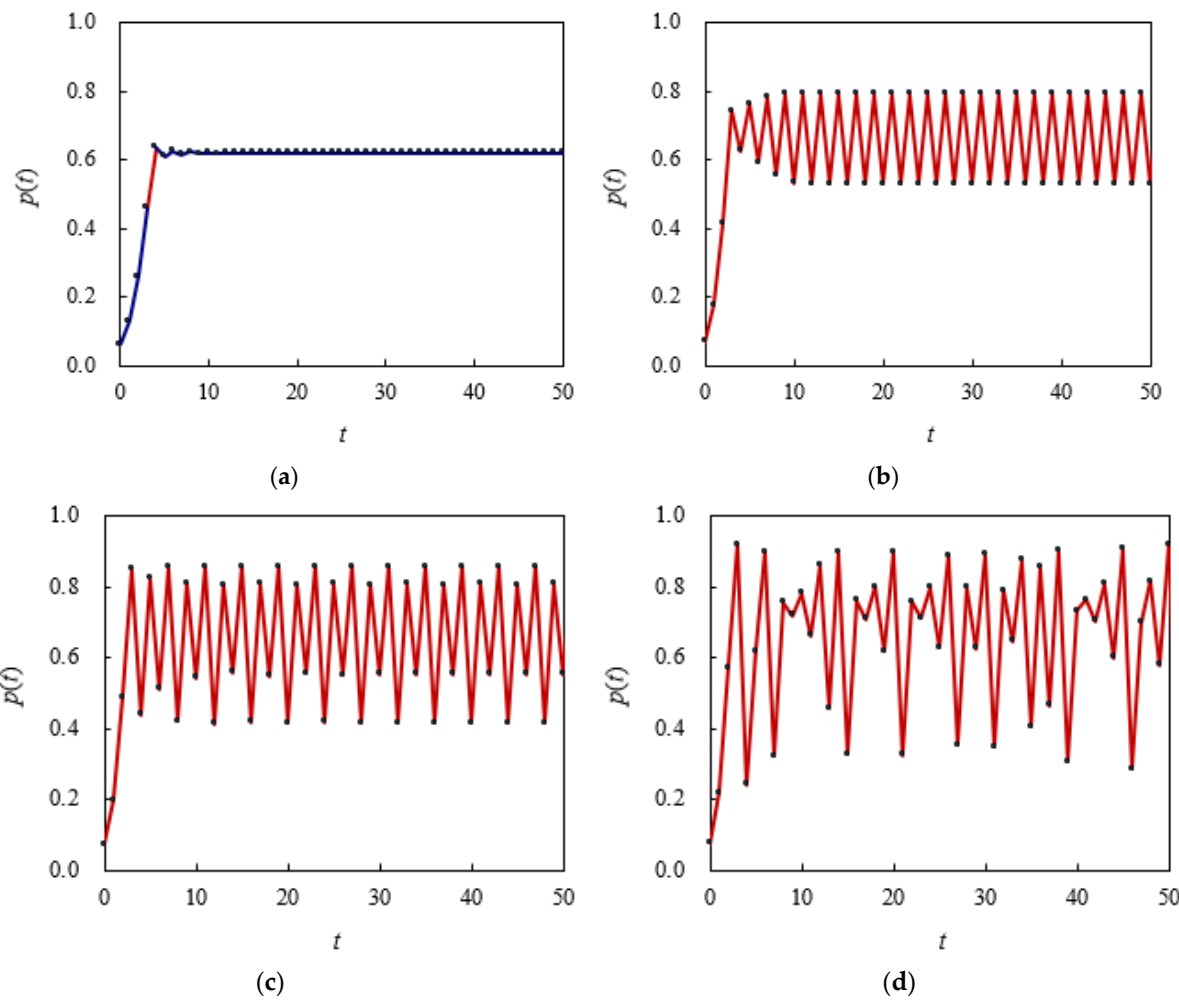

**Figure 5.** Four type of curves indicating nonlinear replacement dynamics: from logistic growth to periodic oscillation, and finally to chaos. This nonlinear dynamic process is a model or metaphor for rapid replacement leading to ikization. The parameter values are as follow: $a = 0.375$, $c = 0.005$, $b = d = 2$, 2.65, 2.95, 3.25. (**a**) Logistic growth, $b = d = 2$; (**b**) Two-period oscillation, $b = d = 2.65$; (**c**) Four-period oscillation, $b = d = 2.95$; (**d**) Chaotic state, $b = d = 3.25$.

## 4. Discussion

Over the past century, many events have happened in China. The tremendous changes in the geographical environment and the repeated setbacks in traditional Chinese culture are both obvious and well-known things. On the other hand, the phenomenon of moral decline in some parts of China raises concerns among knowledgeable individuals. Whether these issues are related to ikization, different people may give different opinions. In scientific research, raising a problem is sometimes more important than solving it. This paper conducts a preliminary study on the modeling and analysis of ikization from the perspective of urbanization. The work has three foundations. The first is a literature review. By reading the literature on Iks, a series of questions worth exploring have been identified. The second is long-term observation of the real world. We have witnessed and heard too many phenomena and facts about cultural breaking and the geographical environment's destruction. The third is analogical analysis. Comparing the social phenomena we have witnessed and heard in the real word the causes and consequences of the Ik people's alienation, and we can link urbanization with ikization. Based on the numerical changes in fractal parameters, empirical analysis of Beijing city demonstrates that the process of urbanization in China is a rapid replacement process. Rapid urbanization can not only lead to the degradation of urban spatial structure, but also cause residents who

migrate from rural areas to urbanized areas to be unable to adapt naturally to the urban environment and culture. Numerical experimentation shows that rapid replacement such as urbanization and natural environmental degradation gives rise to periodic oscillation or even chaos. Oscillation means instability, while chaos means disorder. Although there may be profound order hidden behind chaos, the visible disorder inevitably has a negative impact on human psychology.

This type of research has been rarely reported in the previous literature. The novelty of this paper rests with three aspects. First, the relationships between ikization and fast urbanization of China are explored. Second, a set of mathematical models of ikization is proposed. Third, the model of replacement dynamics for urbanization is associated with the phenomena of ikization. The chief shortcomings of this study are as below: First, lack of systematic historical data. It is impossible to estimate the parameters of Equations (1) and (2), and thus it is difficult to determine the primary and secondary importance of geographical and cultural factors in the process of ikization. Second, lack of systematic field investigation into the real society. Conducting a questionnaire survey and interview analysis on the demolition population caused by rapid urbanization can obtain first-hand information from the real world. Based on questionnaire data, it is expected to discover macro-statistical laws, and interview data can help us reveal the micro-level mechanisms behind statistical laws. Where ikization research is concerned, investigating urban demolition households and reservoir resettlement people is a good choice. This type of work is left for further research in the future.

## 5. Conclusions

The main points of this study can be summarized as follows. First, the replacement dynamics can be employed to model the ikization resulting from geographical processes such as urbanization. Replacement processes can be divided into three groups: natural replacement, rapid replacement, and step replacement. The natural replacement model can be used to describe European and American urbanization, the rapid replacement model can be used to reflect China's urbanization, and step replacement can be used to depict large-scale dismantlement of houses and movement of local residents. It is the step replacement rather than the natural replacement that results in ikization of people in a region or subregions. Second, the strength of the coupling relationship determines the consequences of geographic system transition. The interaction between man and geographical environment, and that between urban populations and rural populations, are nonlinear relationships. If the speed of nonlinear replacement is too fast, it will lead to periodic oscillation and even chaos. Long-term unstable or even disorderly processes inevitably affect human physical and mental health. In this sense, chaos theory may be employed to further interpret the mechanism of ikization. Third, urbanization results in rapid change in the geographical environment and original culture. Because of migration from rural places to urban places, people's living environment will be relatively changed, and meanwhile the rural culture will be suddenly substituted by urban culture. Where the urbanized population is concerned, the original sense of place will disappear overnight. This sudden change may have a subtle negative impact on people's physical and mental health.

**Supplementary Materials:** The following supporting information can be downloaded at: https://www.mdpi.com/article/10.3390/su15129622/s1, File S1: Datasets and sigmoid modeling of fractal dimension of Beijing city (Excel); File S2: Numerical experiments for nonlinear replacement dynamics of urbanization and ikization (Excel).

**Author Contributions:** Conceptualization, Y.C. and L.Y.; methodology, Y.C.; software, Y.L.; validation, L.Y. and Y.C.; formal analysis, Y.C.; investigation, L.Y.; resources, Y.C.; data curation, Y.L.; writing—original draft preparation, Y.C.; writing—review and editing, L.Y.; visualization, Y.C.; supervision, L.Y.; project administration, L.Y.; funding acquisition, Y.C. All authors have read and agreed to the published version of the manuscript.

**Funding:** This research was funded by the National Natural Science Foundation of China, grant number 42171192.

**Institutional Review Board Statement:** Not applicable.

**Informed Consent Statement:** Not applicable.

**Data Availability Statement:** All the datasets appear in the Supplementary Materials of this paper.

**Acknowledgments:** I would like to thank Xingye Tan, for providing remote sensing image interpretation data of urban form of Beijing. Many thanks to anonymous referees whose constructive suggestions help improve the quality of this paper.

**Conflicts of Interest:** The authors declare no conflict of interest. The funders had no role in the design of the study; in the collection, analyses, or interpretation of data; in the writing of the manuscript; or in the decision to publish the results.

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
