# Peer review of "Exploring the Relationship between Urbanization and Ikization"

_sustainability, doi:10.3390/su15129622_

Round 1
Reviewer 1 Report
The paper "Exploring the Relationship between Urbanization and Ikization" presented to me for evaluation is an attempt at contemporary description and connection of the phenomena of urbanization and ikization. The topic is certainly important from the point of view of demographic changes in the world's population and their movement from rural areas to larger urban agglomerations.
In this paper, there is no division into chapters (except for the introduction) such as methodology, results and discussion. Should this paper be treated as a literature review?
I have doubts about the innovative nature of the paper described by the authors in line [59], since all the figures and tables come from an earlier paper, probably by the same author, which can be seen on the following page. In addition, neither in the text nor in the description of the tables and figures there is a reference to those from the following publication:
https://www.semanticscholar.org/paper/Urbanization%2C-Ikization%2C-and-Replacement-Dynamics-Chen/3ca3d18994dbfb302c9c18ffca908c7d56c0156e
Additional notes below: [47-59] - this fragment contains content that should be placed in the executive summary, summary and methodology, and not here. [59-62] – I understand that this is the purpose of the paper? If so, it needs to be described more clearly. [63] – chapter 2 should be a continuation of the introduction. [81-83] - the quote should be written in italics as it is in lines [72-79]. [102, 107, 256, 261, 274, 286, 304, 348, 358, 370, 375] - all formulas 1-11 are written in large font. [138-139] - why are the first and last rows of the table the same? Last to delete. [160-161] - Figure 1 should be placed below the description referenced in the text in line [179]. [234] - Figure 2 - there is Urbanizatio, and there should be urbanization. [315] - Chapter 4.3 should be included as Results, because the authors make considerations on this basis. In addition, part of this chapter should be placed as Methodology, research description. [415 - 446] - The summary is basically a collection of what is already included in the introduction of the paper. There are no strongly articulated conclusions, just a general view. It must be significantly limited and emphasize what is so new in these studies, since the authors write about innovation [59-62]. [448] - Author Contributions: Which author made: software, L.L.? There are only two authors, L.Y. and Y.C. Is it a third? [464-544] - References: · The publication year should be written BOLD · The year of the journal should not be placed in parentheses. · No dot at the end of papers number: 4, 8-10, 12, 22, 23, 25, 26, 30-38, 44-49. · No comma after authors' names.· There should be a diameter between the names of successive authors, not a comma.
Author Response
Please see the attahed file.

Reviewer 2 Report
The article is an interesting take on the subject.
Such analyzes are needed and can add value to science.
The article is written in a correct way, the concept, literature review, description of methodological assumptions and analysis of research results are made in accordance with the standards adopted for this type of studies.
After entering the reviewer's comments, the article may be published.
Issues to be completed:
1. Add the "Materials and Methods" section, which will describe the research objectives, research problems, methods used, list and description of variables, as well as the organization and course of research.
2. Add a "Discussion" part, in which the results will be compared with analyzes of other authors, it will allow to start a scientific discussion.
Reviewer 3 Report
I have regrettably to highlight that the degree of coverage of the submitted manuscript w.r.t. a self-archived paper has emerged as a true concern during my reviewing duty.
Therefore I have spent extra efforts, requiring more days to be allocated in my packed agenda, to duly check if any novelty was actually included in the manuscript to be fair to both authors and prospective readers. Let me report transparently here also to You as authors, and not only as a reserved comment for the editor.
Please, find in the following the findings out of this comparison that make superfluous any mention at the present time of my remarks about the suitability of the submission for the Sustainability journal as well as the quality of the manuscript.
The structure, the contents, the figures, the tables and even the references in the manuscript (made only a couple of exceptions related to papers published after 2015 listed in the following) are perfectly matching the ones in the manuscript uploaded to ArXiv on 4 November 2015:
https://arxiv.org/abs/1511.01180 [checked by pdf version: https://arxiv.org/ftp/arxiv/papers/1511/1511.01180.pdf]
I am aware that self-archiving is a legit practice and does not automatically hinder the possibility of later submitting the manuscript, but the fact that:
1) No mention is made that the manuscript is an extended/updated version of a paper presented or submitted elsewhere; even if a pointer to the archived version is available on the ResearchGate profile of the corresponding author (note: without any reference to the other author);
2) only Yanguang Chen is listed as an author of the manuscript on ArXiv:
Urbanization, Ikization, and Replacement Dynamics
Yanguang Chen
(Department of Geography, College of Urban and Environmental Sciences, Peking University, Beijing,
100871, China)
whilst active roles and responsibilities are credited to the other author signing the current manuscript:
Author Contributions: Conceptualization, Y.C. and L.Y.; methodology, Y.C.; software, L.L.; valida- 448
tion, L.Y. and Y.C.; formal analysis, Y.C.; investigation, L.Y.; resources, Y.C.; data curation, Y.C.; 449
writing—original draft preparation, Y.C.; writing—review and editing, L.Y.; visualization, Y.C.; su- 450
pervision, L.Y.; project administration, L.Y.; funding acquisition, Y.C. All authors have read and 451
agreed to the published version of the manuscript. 452
About data: I have thoroughly checked that no update in the data appears in the submitted manuscript, even if the Statement about
Data Availability incorrectly claims on line 458:
All the datasets appear in the text of this paper.
The only data supplied in the manuscript are included in Table 4 starting at line 394, and by no means allows for reconstructing raw data behind the calculations and estimates supplied in the text, indeed.
These practices do not appear to be consistent with the authors' commitment to publication ethics, research ethics, copyright, and authorship guidelines published on the journal web pages.
The actually modified sentences are:
- In Section 2.2
In recent years, the 134 Chinese government has finally realized the importance of protecting the geographical 135
environment. A series of new policies and measures have been introduced, which have 136
significantly improved China's natural environment and ecological conditions. 137
In modern society, the source of political corruption is the very source of unsustain- 163
able development. [...] 164
- In Section 3
In geography, urbanization involves both quality and quantity. [...] 199
Both urban system and urban form can be well 201
described from the prospective of natural cities [32, 33]. However, it is difficult to study 202
urban ecology and urbanism through modern technology. [...] 203
- In Section 4.3
Because of the destruction of geographical environment and the rupture of tradi- 316
tional culture, [...] 317
On lines 386-388 the mention that the fractal dimension growth of Beijing
can be modeled by a quadratic logistic func- 387 tion [49].
is set by referencing:
Chen YG. Logistic models of fractal dimension growth of urban morphology. Fractals, 2018, 26(3): 1850033 543
whilst in the original manuscript it is reported by another equation (numeberd 12) which reports exactly the same model in the above mentioned paper (on page of ArXiv copy below eqaution 10 in the Accepted Manuscript).
- New references:
Chen YG. Logistic models of fractal dimension growth of urban morphology. Fractals, 2018, 26(3): 1850033 on line 543
Jiang B, Miao Y. The evolution of natural cities from the perspective of location-based social media. The Professional Geographer, 2015, 67(2): 295–306 on lines 516-517
Jiang B, Yin J, Liu Q. Zipf’s law for all the natural cities around the world. International Journal of Geographical Information Science, 2015, 29(3): 498–522 on lines 518-519
Reviewer 4 Report
Check the PDF copy of the article for correction.

Round 2
Reviewer 1 Report
Dear Editors and Authors,
Thank you for making the corrections. I propose to submit publications for printing.